# A Framework for Generating Dangerous Scenes for Testing Robustness

**Shengjie Xu**
University of California, Santa Cruz
sxu88@ucsc.edu

**Lan Mi**
University of California, Santa Cruz
lmi6@ucsc.edu

**Leilani H. Gilpin**
University of California, Santa Cruz
lgilpin@ucsc.edu

## Abstract

Benchmark datasets for autonomous driving, such as KITTI, nuScenes, Argoverse, or Waymo are realistic but designed to be faultless. These datasets contain limited errors, difficult driving maneuvers, or other corner cases. We propose a framework for perturbing autonomous vehicle datasets, the DANGER framework, which generates edge-case images on top of current autonomous driving datasets. The input to DANGER are photorealistic datasets from real driving scenarios. We present the DANGER algorithm for vehicle position manipulation and the interface towards the renderer module, and present five scenario-level dangerous primitives generation applied to the virtual KITTI and virtual KITTI 2 datasets. Our experiments prove that DANGER can be used as a framework for expanding the current datasets to cover generative while realistic and anomalous corner cases.

## 1   Introduction

Since the release of the KITTI [Geiger et al., 2012] dataset, autonomous driving research has been data-driven. Autonomous vehicles (AVs) promise to decrease vehicle fatalities and increase safety in the modern automobile. However, the majority of datasets [Chang et al., 2019, Wilson et al., 2021, Sun et al., 2020, Caesar et al., 2020, Geiger et al., 2012] and derived algorithms [Cheng et al., 2020, Xu et al., 2022, Girshick, 2015, Ren et al., 2015, Reading et al., 2021, Ganeshan et al., 2021, Cai et al., 2021, Weng et al., 2020, Zhou et al., 2020] are used for benchmarking and standardized on perfectly curated datasets, This causes two issues: (a) these algorithms are specially designed and hard-code into a workable scenario (b) solutions are focused on accuracy on a single dataset, rather than robustness. Instead, we want autonomous driving solutions to be able to deal with different driving scenarios. A real road scene is often dangerous and full of all kinds of unexpected events. There is an immediate need for AI systems to consider these event, especially in the context of implementing algorithms on actual on-road vehicles [Shalev-Shwartz et al., 2017]. Instead, we propose the development of a framework to mimic the kinds of "one-off" scenarios humans may encounter in driving tests. We use this to validate that AI systems can generalize in real-world driving environments. We describe an iterative procedure for creating out-of-domain examples for autonomous driving, or corner cases, based on the existing AV standard datasets.

It is challenging to design benchmarks and datasets. Therefore, we contribute a general framework for generating photo-level realistic driving scenes with custom trajectory inputs. Our framework, "DANGER", supports user-defined vehicle trajectories and poses to complete a sequence of frames of data generation. For example, a user can define a left-turn maneuver, which can be included in the

data set. DANGER also supports the distortion and deletion of vehicles in an individual frame and can simulate illogical special camera failure modes. In summary, our key contributions are:

- We introduce DANGER, a framework for the generation of Danger-Aware datasets. DANGER is a generator that can enhance robustness. It generates new scenarios with user input: a set of primitives. Each primitive is a vehicle driving trajectory and posture over time: a complete sequence of frames of data generation.
- DANGER supports the shifting and deletion of cars in individual frame and can simulate illogical special camera failure modes.
- Our DANGER implementation includes five scenario-level dangerous primitives applied on virtual KITTI and virtual KITTI 2 to generate more robust, "DANGER-vKITTI" datasets.
- We evaluate DANGER on a corner case score evaluation and via human study. Our results demonstrate that DANGER and our generated scenarios are realistic, novel, anomalous, or risky. These dataset augmentations can help increase the robustness and range of scenarios in the original datasets.

To prove that DANGER that can generate anomalous driving scenarios, we implemented quantitative and qualitative experiments on two datasets: Virtual KITTI [Gaidon et al., 2016b] and Virtual KITTI 2 [Cabon et al., 2020]. Our code for the benchmarks and results of experiments have been open-sourced under the MIT License. It is available at: `https://github.com/link` (Link is hidden for double-blind review).

## 2 Related Work

Today's deep neural networks are highly dependent on goodness-of-data and obey the "garbage in, garbage out" rule of thumb [Vidgen and Derczynski, 2020, Kim et al., 2016]. As Sambasivan et al. [2021] suggested, there is no goodness-of-fit without goodness-of-data. In the absence of standardized metrics to characterize "the goodness" of data, conventional visual perception methods are often not able to detect dangerous scenarios. This is because corner cases have not been witnessed during training [Thomas and Uminsky, 2020, Breitenstein et al., 2021]. However, fitting metrics does not represent the phenomenological fidelity and validity of the data. Their detection is based on sanitized data, lacking anomalous events: low possibility but realistic dangerous driving scenarios. Therefore, we need reliable detection and understanding of corner cases, which will further increase safety in autonomous driving.

### 2.1 Autonomous Driving Datasets

KITTI [Geiger et al., 2012] is the pioneering benchmark datasets for use in autonomous driving by providing LiDAR sensors, stereo cameras, and GPS/IMU data. Compared with KITTI, Waymo [Sun et al., 2020] provides a large-scale, high-quality dataset with high-intensity annotations and higher annotation frequency using five cameras and five LiDARs from different angles and locations. nuScenes [Caesar et al., 2020] provides 360°coverage from the LiDAR, radar, and camera sensors. Argoverse [Chang et al., 2019] contributes detailed geometric and semantic maps of the environment, and its sibling Argoverse2 [Wilson et al., 2021] has the most extensive self-driving taxonomy with HD maps that include real-world changes. Cityscapes [Cordts et al., 2016] provides semantic, instance-wise annotations for semantic understanding of urban street scenes. The current state-of-the-art (SOTA) datasets are inherently designed for independent model training based on various sensors rather than handling real-world challenges. As Shalev-Shwartz et al. [2017] estimated that AVs need to drive 30 billion miles to get enough statistical evidence to prove that AVs are three times safer than human drivers, yet leading Waymo claims their test fleet has run 20+ million miles on public roads [Schwall et al., 2020]. Therefore, corner case or out-of-distribution data in existing autonomous driving datasets are insufficient.

### 2.2 Synthetic Datasets

Advances in computer graphics have made it possible to easily annotate and generate virtual datasets, such as SYNTHIA, Virtual KITTI (vKITTI), and Virtual KITTI 2 (vKITTI2), include various scene types under different weather, environment, and lighting conditions [Ros et al., 2016, Gaidon et al.,

2016b, Cabon et al., 2020]. Johnson-Roberson et al. [2016] demonstrate that SOTA neural networks trained using only synthetic data perform better than the same architectures trained on real-world dataset. CARLA [Dosovitskiy et al., 2017] supports custom waypoints input for vehicle trajectory generation, but it is projected in a virtual city built in a game engine without any modification compatibility to an existing real-world dataset.

## 2.3 2D Image Synthesis

Generative Adversarial Networks (GANs) [Goodfellow et al., 2014], have been used for a variety of image synthesis exercises, including image generation [Radford et al., 2015, Arjovsky et al., 2017, Karras et al., 2017], image-to-image translation [Isola et al., 2017, Zhu et al., 2017], text-to-image synthesis [Zhang et al., 2017, Reed et al., 2016], and inpainting [Pathak et al., 2016]. StyleGAN [Karras et al., 2019a] generates high-quality, high-resolution face images. StyleGan can also generate car-like images; however, flaws remain in the generated dataset. Firstly, the generated images were frequently displayed at a 45-degree exhibition angle rather than the perspective view of a car in motion. Second, the shape of some of the cars was distorted and produced a peculiar effect of indistinguishable front and rear of the car. StyleGAN2 [Karras et al., 2019b] fixed the artifacts problem remained in StyleGAN. However, both StyleGAN and StyleGAN2 cannot generate images via controlling car object-dependent appearance, pose, and size in 3D.

Recent studies [Ramesh et al., 2021, 2022, Saharia et al., 2022] showed the SOTA GPT-based methods of generating photorealistic images from text. Nevertheless, the above methods generate images that have low fidelity and cannot generate continuous frame images or videos.

Several studies [Ratner et al., 2017, Cubuk et al., 2019] learned transformation and color adjustment policies, such as rotate, shear, shear, contrast, or, hue. These strategies, however, are limited to 2D images or color space and are challenging to generate new viewpoint data. Contrary to 2D, we need a disentangled 3D-aware image synthesis model that allows a user to edit the viewpoint, object shape, or texture independently [Zhu et al., 2018].

## 2.4 3D-aware Image Synthesis

Several recent studies [Yao et al., 2018, Schwarz et al., 2020, Liao et al., 2020] developed multiple GAN-based 3D-aware image synthesis methods including 3D-friendly features: interpretable, disentangled scene representation, viewpoint manipulation, and 3D Controllable. Wu et al. [2017], Liao et al. [2020], and Greff et al. [2022] inspired by rendering a 2D image by 3D game engine, which treats images as a projection of the 3D world. The learned 3D space can be potentially useful for various tasks such as image reasoning tasks in the dangerous scene understanding.

Inspired by Mildenhall et al. [2020], GRAF [Schwarz et al., 2020] introduces GANs to implement Neural Radiance Fields, and uses conditional GAN [Mirza and Osindero, 2014] to achieve controllability of the rendered object. GIRAFFE [Niemeyer and Geiger, 2021] uses one Neural Radiance Field per object in order to combine objects from different scenes. This enables the movement and rotation of objects in the generative new image. GRAF and Liao et al. [2020] can generate images of cars in different poses, but the resulting backgrounds are often monochromatic or pure white. 3D-SDN [Yao et al., 2018] and PNF [Kundu et al., 2022], are the algorithm that can modify both the 3D pose and position of the spawning vehicles while remaining realistic city and road scenes.

## 2.5 Robustness

Our framework is a hybrid knowledge representation layer that improves robustness by introducing out-of-domain semantically-significant samples. Our work is similar to a knowledge-driven cognitive model approach improve AI system robustness [Marcus, 2020], with a focus on robustifying autonomous vehicle data sets with predefined primitives. We argue that using a primitive representation, similar to the abstract script-like representation in language [Schank, 1975, Borchardt, 1992], can make the underlying opaque system more understandable. Amini et al. [2022] demonstrates that VISTA simulator can generate novel camera images. However, the generated photo-realistic pictures can only be obtained via changing the driving angle of the ego vehicle but not the pose of other cars on the road.

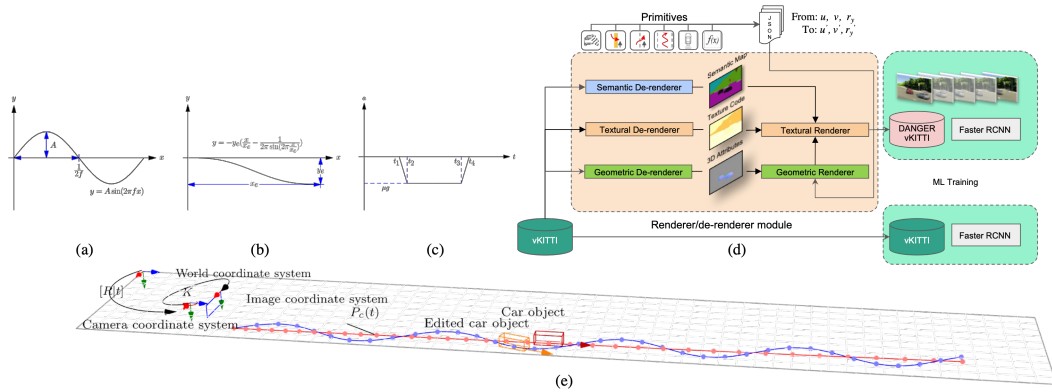

Figure 1: **DANGER framework, primitives, and visualization of scene editing**. (a) Primitive slalom lane change function (b) cut-in function (c) braking function (d) DANGER contains a renderer/de-renderer module, primitive function module, and generated descriptive file that can help users to develop a wide variety of corner-cases (e) a slalom-lane-change scene editing of scene `0006` shown in world coordinate. The orange red curve is the original trajectory of car object `tid` 4, and the light blue curve is the corresponding edited trajectory in world coordinate.

Our work is inspired by prior work on a stress testing framework for autonomous system verification and validation [Falco and Gilpin, 2021]. The key idea for DANGER is that the robustness is not static, but a series of stress tests that can be automatically generated. Our goal is to use these generative examples to generate counterfactuals, similar to previous work using adversarial examples for explaining counterfactuals [Pawelczyk et al., 2022]. Our contribution is a set of semantic primitives, similar to the semantic layer in DNNs discussed in [Browne and Swift, 2020].

According to the systematization of corner case detection complexity proposed by Breitenstein et al. [2021], our framework is designed to obtain the most dangerous anomalous scenario for a detector, and we also answer the research question, '*How to generate or record corner case from descriptions?*', proposed by Bogdoll et al. [2021]. Breitenstein et al. [2021] defined a corner case as a non-predictable relevant object/class in a relevant location, Breitenstein et al. [2021] extended the definition of corner case level by giving three most dangerous scenes: the anomalous scenario is a potentially dangerous unknown object, the novel scenario is a harmless unknown object, the risky scenario is a potentially dangerous but known object. In the result section, we present the design our primitives.

## 3 Method

**Renderer/de-renderer module**    The 3D scene de-rendering networks (3D-SDN) [Yao et al., 2018] is an optimal algorithm that generates photo-level realistic synthetic images. It employs an encoder-decoder architecture with three branches: scene semantics, object geometry, 3D pose, and textual appearance of objects and the background. As shown in Figure 1 (d), three branches intend to learn a scene's semantic segmentation, infer the object shape and 3D pose, and encode the appearance of each object and background segment. Disentangling 3D geometry and pose from the given scene enables 3D-aware scene manipulation in image coordinate with a target object location $(u, v)$, orientation pose $r_y$, and (`delete`, `modify`) operations described in a JSON file.

In the result section, we use 3D-SDN as the renderer/de-renderer module to demonstrate the feasibility of our framework. In practice, modules such as PNF can also be selected [Kundu et al., 2022].

**Primitives**    To augment the input dataset, we define five danger-aware primitives: `Exit parking`, `Cut-in Opposite`, `Cut-in`, `Slalom Lane Change`, and `Braking`. Detail of these primitives will be introduced in the results section.

### 3.1 Scene Editing Computation

Our approach is adaptive. The object's position can be edited in the real-world 2D plane according to any arbitrary function. At the same time, the implementation of editing in the world coordinate system requires reading the position information of the object and transforming it into the world coordinate system. The vKITTI dataset was designed to match the multi-object tracking (MOT) evaluation benchmark of KITTI. Therefore, the MOT ground truth and exact position of each car object are provided in the folder `motgt`. The following object annotation and terminology we inherited from vKITTI is detailed in Gaidon et al. [2016b,a] and supplementary material.

We assume that the original driving trajectory of the target vehicle object is a straight line. Given a image $\mathcal{I} \in \mathbb{R}^{W \times H \times 3}$ defined by `scene`, `topic`, `tid`[1], and `frame` with known camera intrinsic matrix $\mathbf{K} \in \mathbb{R}^{3 \times 4}$ and camera extrinsic matrix $[\mathbf{R}|\mathbf{t}]$, we can convert the position of an object in world coordinate, camera coordinate, or pixel coordinate by Equation (1) and Equation (3), where $\mathbf{R} \in \mathbb{R}^{3 \times 3}$ and $\mathbf{t} \in \mathbb{R}^{3 \times 1}$ indicate the rotation and translation matrices [Hartley and Zisserman, 2003]. $P_c \in \mathbb{R}^{4 \times 1}$ is the 3D point position in camera coordinates and $P_w \in \mathbb{R}^{4 \times 1}$ is the 3D point position in world coordinates. In both cases, they are represented in homogeneous coordinates. A camera extrinsic matrix $\mathbf{M} \in \mathbb{R}^{4 \times 4}$ is used to denote a projective mapping from world coordinates to pixel coordinates.

$$z_c \begin{bmatrix} u \\ v \\ 1 \end{bmatrix} = \mathbf{K} \begin{bmatrix} \mathbf{R} & \mathbf{t} \\ \mathbf{0} & 1 \end{bmatrix} \begin{bmatrix} x_w \\ y_w \\ z_w \\ 1 \end{bmatrix} = \mathbf{K} \begin{bmatrix} x_c \\ y_c \\ z_c \\ 1 \end{bmatrix} = \mathbf{P} \begin{bmatrix} x_w \\ y_w \\ z_w \\ 1 \end{bmatrix} \tag{1}$$

$$\mathbf{P} = \mathbf{KM} \tag{2}$$

$$P_c = \mathbf{M} P_w \tag{3}$$

The elements of object's center position vector $P_c$ can be acquired from the MOT ground truth data `x3d`, `y3d`, `z3d`, and `h3d`, and the corresponding $P_w$ will be easily obtained by apply the inverse of frame-dependent matrix $\mathbf{M}$. In the x-z plane, arbitrary vehicle poses can be generated according to the primitive function, where $\mathbf{r}'_y$ is a unit tangent vector to the curve at $(x'_w, z'_w)$ representing the target orientation of the car object. The trajectory curve and heading vectors are generated at the origin and then translated to the desired starting point. The curve was further rotated by $\theta$ using the rotation matrix $\mathcal{R}_\theta$ to align with the original trajectory's slope $a$, where $a$, $\theta$ are the slope and angle of original path, $\mathcal{R}$ is a corresponding rotation matrix [Weisstein, 2003]. A detailed algorithm of the implementation of the primitive operation in 3D world coordinate is shown in the supplementary material.

## 4 Results

In this section, we present the *base* dataset and primitives we used in our generation framework. We evaluate our results in two aspects that focus on the corner cases generation capability and the realistic level of our dangerous maneuver via a human study.

We hypothesize that adding risky maneuvers to the dataset will result in a higher corner case score for the ego vehicle. We performed a numerical analysis of our generated data frames by applying a corner case detector that considers object-level and predictability. We also validated our results with a user study.

### 4.1 Dangerous Corner Case Generation

#### 4.1.1 Primitives

We designed five scenario-level dangerous corner cases in the world space based on the vKITTI dataset. These primitives are deliberately selected for each scene presented in the vKITTI according

---

[1]A unique track identification number for each object instance in the scene.

to the vehicle object's position and motion. Though these are hand-curated, each primitive follows the definition of a scenario-level corner case that is an anomalous or risky scenario derived from the real-world. We also refer to the safety assist test procedure and autonomous vehicle collision report as templates for our dangerous corner cases [Van Ratingen, 2017, California DMV, 2022e].

**Cut-in** Many accidents are caused by neighboring vehicles suddenly driving in front of a moving car dangerously, either due to the driver impatiently overtaking or an unintentional aggressively traverse due to forgetting the highway exits. To achieve a realistic cut-in lane change, we define the single-lane change curvature according to the sinusoidal ramp function [Sledge Jr and Marshek, 1997] as following:

$$y = y_e \left( \frac{x}{x_e} - \frac{1}{2\pi \sin(2\pi \frac{x}{x_e})} \right), \tag{4}$$

where $x_e$ is the longitudinal offset of the target position, $y_e$ is the lateral lane-change offset of the target position as shown in Figure 1 (b). The forwarding trajectory will be rotated and aligned with the camera space's longitudinal axis $z_c$. We chose two vehicles with `tid` 1 and 2 in scene `0018` to simulate two scenarios of cut-in ego vehicle and overtaking with nine sets of parameters, respectively.

**Exit Parking** A careless driver may suddenly place its front end out of a line of parked cars on either side of a narrow street. We assign a trajectory generated by the cut-in function presented above to two distinguished car objects with `tid` 63 and 70 in scene `0001`. Among the nine sets of parameters for each vehicle, $x_e$ was chosen as a distance of 4 m for one and a half vehicle lengths.

**Cut-in on Opposite** We incorporate the rotated cut-in function to a car (`tid` 0) driving in the opposite direction. The rampage driver's driving leads to an upcoming accident that potentially causes severe injury in scene `0002`. By combining six $x_e$ and three $y_e$ parameters, we obtained eighteen sets of different driving conditions distributed in a two-dimensional space of ninety square meters.

**Slalom Lane Change** According to Wang [1994]'s study, lane change crashes caused over 244,000 accidents in 1991, accounting for 4.0% of all accidents in the US. Therefore, we aim to design a novel scenario that a driver can barely decide whether other car objects choose to make a lane change or not. As illustrated in Figure 1 (a), we borrow the idea from the slalom test in automotive engineering and assign a parameter-dependent sine-wave to the `tid` 2 and 7 in scene `0006` as follows:

$$y = A sin(2\pi f x) \tag{5}$$

where $A$ is the lateral offset amplitude in meters, and $f$ is the steering input frequency in Hz.

**Braking** We simulate a constant deceleration maneuver in scene `0020` when `tid` 16 applies a braking pedal by exploiting the inverted trapezoid piecewise function:

$$a_t = \begin{cases} 0, & t < t_1 \\ -\frac{(t-t_1)\mu g}{t_2-t_1}, & t_1 \le t < t_2 \\ -\mu g, & t_2 \le t < t_3 \\ -\frac{(t_4-t)\mu g}{t_4-t_3}, & t_3 \le t < t_4 \\ 0, & t_4 \le t \end{cases} \tag{6}$$

where $a_t$ is the time dependent acceleration of the editing vehicle, $\mu$ and $g$ constitute a uniform deceleration value, $t_i$ is the timestamp when the braking caliper initiated and stopped. As illustrated in Figure 1 (c), we replace the object speed in the original video frame with the target braking speed curve obtained by filtering and integrating the generated acceleration curve. Then a new waypoint can be generated in world coordinates to replace the original.

For each primitive, we also calculate the rotation $\Delta r_y$, zoomed-in factor $\rho$, and the transformed $(u, v)$ pixels in camera space. All operation parameters are packaged into a JSON file for 3D-SDN processing. For the detailed JSON and parameter settings, see our supplementary material.

### 4.1.2 Datasets

vKITTI and vKITTI2 [Gaidon et al., 2016b, Cabon et al., 2020] contain five scenes of photo-realistic videos synthesized from virtual worlds under various lighting and weather conditions, respectively. vKITTI2 has more photo-realistic images by utilizing the updated Unity game engine enhancements.

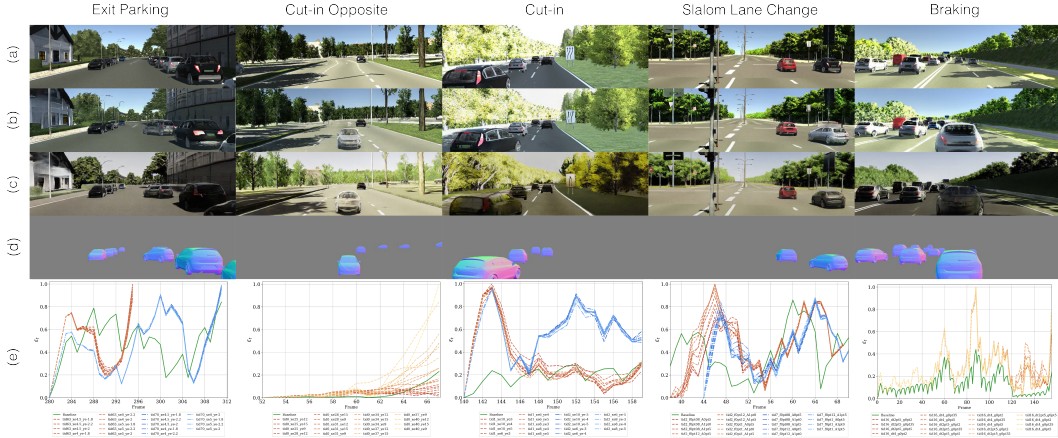

Figure 2: **Primitive dangerous scenarios and evaluation results**. Generation result for each primitive in our study from left to right: `Exit parking` (frame301), `Cut-in Opposite` (frame67), `Cut-in` (frame144), `Lane Change` (frame52), and `Braking` (frame129). The (a) vKITTI, (b) DANGER-vKITTI, (c) DANGER-vKITTI2, (d) normal map, and (e) a normalized dangerous corner scores evaluation of the generated DANGER datasets (two `tids` in red and blue color cluster, baseline vKITTI in green) are shown from top to bottom.

| No. | Metrics | Datasets | Exit Parking | Cut-in Opposite | Cut-in | Slalom Lane Change | Braking |
|---|---|---|---|---|---|---|---|
| (1) | $\bar{\epsilon}$ | vKITTI | 0.48 | 0.04 | 0.19 | 0.41 | 0.13 |
| (2) | $\epsilon^*$ | vKITTI | 0.84 | 0.23 | 0.31 | 0.86 | 0.44 |
| (3) | avg. of $\bar{\epsilon}_{\mu \in \mathcal{D}}$ | ours | 0.46 | 0.07 | 0.44 | 0.45 | 0.26 |
| (3)/(1) | Ratio | | 0.97 | **1.97** | **2.27** | **1.09** | **1.99** |
| (4) | avg. of $\epsilon^*_{\mu \in \mathcal{D}}$ | ours | 0.95 | 0.29 | 0.96 | 0.89 | 0.98 |
| (4)/(2) | Ratio | | **1.13** | **1.26** | **3.15** | **1.03** | **2.23** |
| (5) | $\epsilon^*_{\mu \in \mathcal{D}}$ | ours | 1.00 | 1.00 | 1.00 | 1.00 | 1.00 |
| (5)/(2) | Ratio | | **1.19** | **4.33** | **3.27** | **1.16** | **1.21** |

Table 1: **Quantitative evaluation of DANGER-vKITTI with the baseline dataset vKITTI**. We compared the mean $\bar{\epsilon}$ and max value $\epsilon^*$ of the corner score for each parameter setting in each scene by its average number and maximum number.

In our experiments, we downscale vKITTI images to $624 \times 192$. Our inference model was inherited from Yao et al. [2018], and we also trained models followed the setup of Yao et al. [2018] for vKITTI2 on an NVIDIA Tesla P100 (16G) GPU.

Our DANGER-vKITTI dataset contains 4527 pictures with consecutive frames for each of the five scenes with 18 different sets of parameters. Compared to the original five scenes from vKITTI, DANGER has expanded the original dataset into 90 additional scenes using the experimental parameters.

### 4.2 Quantitative Evaluation

We use the conceptual definition of the corner case Bolte et al. [2019] proposed that a *non-predictable relevant object/class* in *relevant location*, and we hypothesize a dangerous movement is a subset of corner cases. Our corner case detector considers semantic segmentation and image non-predictability.

First, we adopt a segmentation network based on the `MobileNet-V2` [Sandler et al., 2018] that allows us to classify and localize the *objects* in the scene for which moving objects are considered as *relevant*. Specifically, we use the model pretrained on ImageNet, MS-COCO, and Cityscapes `train_fine` set provided in `DeepLabv3` [Deng et al., 2009, Lin et al., 2014, Cordts et al., 2016, Chen et al., 2017]. Second, we used the advantage of `PredNet` [Lotter et al., 2016] that can sensing

the moving objects to compare the deviation between predicted frame $\hat{\mathbf{x}}_t$ and the real next frame $\mathbf{x}_t$ for time $t$. As a metric for the corner case, we calculate an error $e_t$ to represent the *non-predictable* component of the detection system.

$$\mathbf{e}_t = \hat{\mathbf{x}}_t - \mathbf{x}_t, \tag{7}$$

where elements $e_t(i), i \in \mathcal{I}$. Here, each pixel $i \in \mathcal{I}$, where $\mathcal{I}$ denotes the set of pixel indices in the given image, and $\|\mathcal{I}\| = H \cdot W$ represents the number of pixels. The input image $\mathbf{x}_t \in \mathbb{G}^{H \times W \times C}$ with image pixel $x_t(i) \in \mathbb{G}$, where $\mathbb{G} = \{0, 1, \ldots, 255\}$ is the set of gray values, $H$ and $W$ are the image height and width in pixels and $\mathbb{C} = \{1, 2, 3\}$ is the number of color channels. The segmentation network maps the input to output scores $\mathbf{P}_t \in \mathbb{I}^{H \times W \times \|\mathcal{S}\|}$, where $\mathcal{S}$ denotes the set of classes with cardinality $\|\mathcal{S}\| = 19$ and $\mathbb{I} = [0, 1]$. Taking the argmax over the output scores we obtain the $(H \times W)$-dimensional mask $\mathbf{m}_t = \arg\max_{s \in \mathcal{S}} \mathbf{P}_t$, which gives us a pixel-wise classification $m_t(i) \in \mathcal{S}$. Within this work, we limit the target objects $s \in \mathcal{S}_{rel} = \{12, 13, \ldots, 19\}$, which represents the *relevant classes* such as `Person`, `Car`, `Truck`, etc [Cordts et al., 2016].

The information from the two preceding processing steps is combined as a detection system as metric $\epsilon$ to evaluate our DANGER datasets. Since we only interested in the target classes defined in $\mathcal{S}_{rel}$, we filter the error map Equation (7) by the following formula:

$$e_{t,rel}(i) = \begin{cases} e_t(i), & m_t(i) \in \mathcal{S}_{rel} \\ 0, & m_t(i) \notin \mathcal{S}_{rel} \end{cases} \tag{8}$$

To assign higher dangerous weighting for the objects close to the ego vehicle, we employ the weighted squared errors of the relevant classes introduced by Bolte et al. [2019] as follows:

$$\epsilon' = \sum_{i \in \mathcal{I}} e^2_{t,rel}(i) \cdot (1 - \frac{h_i}{H - 1}), \tag{9}$$

with $h_i \in \{0, 1, \ldots, H - 1\}$ being the row index from bottom-up. In contrast to Bolte et al. [2019], we normalize the error scores $\epsilon'_t$ of $m^{\text{th}}$ dangerous parameter in set $\mathcal{D} = \{1, 2, \ldots, 18\}$ for the same scene to ensure the peak dangerous frame $\mathbf{x}_t$ of the system generates a corner case score $\epsilon_t = 1$. The corner case score is obtained by normalizing the error score $\epsilon'_t$ to a value range from 0 to 1 using:

$$\epsilon_t = \frac{\epsilon'_t - \min_{\tau \in \mathcal{T}, \mu \in \mathcal{D}} \epsilon'_{\tau,\mu}}{\max_{\tau \in \mathcal{T}, \mu \in \mathcal{D}} \epsilon'_{\tau,\mu} - \min_{\tau \in \mathcal{T}, \mu \in \mathcal{D}} \epsilon'_{\tau,\mu}} \tag{10}$$

where $\mathcal{T}$ denotes a set of timestamps, and $\mu$ denotes the parameter set in $\mathcal{D}$.

Compared to the green baseline datasets, Figure 2 (e) shows that the corner score increases regardless of which `tid` object is modified. The highest score for `Exit parking` is in the last frame (296 and 312) when the car is completely exiting the side parking; the highest score for `Cut-in Opposite` is in frame 67 when the vehicle is about to crash into the ego; the dangerous score of the `Cut-in` is highest in frame 143 when the black car starts to leave the lane and in frame 152 when the gray car is heading to the opposite lane; the dangerous score is higher in the `Slalom Lane Change` between frames 43 and 52 when the twisting vehicle is relatively close; in the `Braking`, the dangerous score is relatively high in the last frame, 158, and the peak in the middle may be caused by the sudden disappearance of the vehicle object in the frame due to the instability of the 3D-SDN model. Table 1 shows that DANGER's corner case level is between 1 and 3.15 times that of the original datasets considering the average performance across parameters. Considering only the individual parameters, DANGER's hazard factor can be up to 4.33 times the original data.

## 4.3 User Study

We designed a user study to validate our synthetically generated data samples, e.g., scenarios. Our hypothesis was that users would find the scenarios realistic and the level of "dangerousness" would correlate with Table 2 and Figure 2 (e). We recruited 100 subjects, who were described the domain and problem setup: that we have augmented a data set to create new scenarios that they are to rank. Users were also presented with the definitions of novel, anomalous, risky and unknown/known events, which are consistent with Breitenstein et al. [2021]. Subjects were recruited via Amazon Mechanical Turk and compensated for completing the survey.

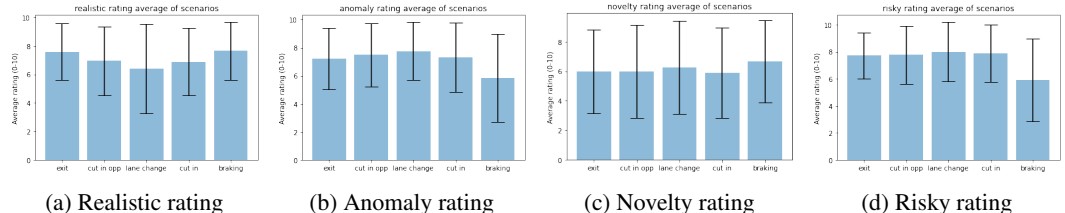

|  | (a) Realistic rating | (b) Anomaly rating | (c) Novelty rating | (d) Risky rating |

Figure 3: Aggregated user study results for rating our scenarios as realistic, anomalous, novel and risky. Averages are plotted as the bars, and error bars show the standard deviation.

| Primitive | Level | Testimony |
|---|---|---|
| Exit Parking | Novel | *'…a parked pickup …pulled out …and made contact with …the Waymo AV…'* [California DMV, 2022c] |
| Cut-in Opposite | Anomalous | *'…a sedan was traveling the wrong way on a one-way section …made contact with the Cruise AV…'* [California DMV, 2022b] |
| Lane Change | Novel | N/A |
| Cut-in | Risky | *'…a passenger vehicle in the right adjacent lane abruptly cut in front of the Zoox vehicle…'* [California DMV, 2022d] |
| Braking | Risky | *'…The passenger vehicle ahead …transitioned into reverse and began to accelerate, making contact with the front sensor of the Waymo AV…'* [California DMV, 2022a] |

Table 2: We classify our primitives under the taxonomy of corner cases defined by Breitenstein et al. [2021]. We also provide descriptive prompts derived from Autonomous Vehicle Collision Reports in 2022 for each primitive [California DMV, 2022e].

Users were presented with a driving scenario and answered four questions for each scenario: to rank whether the scenario was realistic, novel, anomalous and risky. Users were presented a Likert scale from 0-10, with 10 indicating very realistic/novel/anomalous/risk, and 0 being not realistic/novel-/anomalous/risk at all. We presented a random sample of scenarios, one per primitive. The questions were presented in a random order. The results are in Figure 3.

We found that users generally agreed with our classifications. Users found our scenarios to be realistic, rating them with average scores in the 7-8 range. Users found the cut-in opposite and lane change scenarios to be the most anomalous. Users found the lane change and braking scenarios to be the most novel. Users found most of the scenarios to be risky. Notably, they did not find the braking maneuvers to be very risky. This might be because the braking scenarios are almost 1 minute in length, and perhaps users did not watch, and therefore analyze, the full scenario.

## 5   Conclusion and Discussion

In this paper, we present DANGER: a framework for generating anomalous driving scenarios from existing self-driving vehicle datasets. We show how to define a set of primitives which align with corner cases [Brazil et al., 2020] and user feedback. We promote the use of DANGER to robustify existing autonomous vehicle datasets that may not contain error cases. In fact, it might be implausible to collect such corner cases, either because of the intractability of the scenario, or because of the ethical consequences. We present a framework and technique to define, extract, and classify these scenarios *from existing data*.

Our work has limitations. The 3D-SDN model may generate a jitter frame due to the geometry estimation failure, this might be improved by choosing a more robust renderer/de-renderer module. Similarly, we sometimes get a missing car in a single frame, which may lead to errors in the calculation of the dangerous score. Currently, we can only complete modifications with 3 degrees of freedom. These limitations highlight the challenge of generating corner cases.

We show that "dangerousness" is a superclass consisting of novel, anomalous and risky behaviors and actions. Our initial set of primitives are intentionally distributed over anomalous, novel, and risky scenarios. Users generally agree with labels and categorizations. While we demonstrated DANGER on vKITTI, it can be used with any autonomous vehicle dataset that supports semantic and 3D annotations with a given renderer.

In summary, our DANGER framework is a robustness generator for self-driving car datasets. It is adaptable to multiple types of primitives, and it can cover a wide range of dangerous levels: novel, risky, and anomalous. Our work has opened a new area of robustness data generation, where users, stakeholders, and system designers can identify and easily generate corner cases to augment datasets in order to make them more robust.

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
