# OpenReview forum: "A Framework for Generating Dangerous Scenes for Testing Robustness"
_NeurIPS.cc/2022/Workshop/TEA — TEA_

### Official Review · Reviewer_B4hT · 2022-10-13
**Promising idea to use interpretable primitives to synthesize dangerous driving scenarios, but results do not sufficiently justify or evaluate the method**

**Rating:** 4
**Confidence:** 3

**Review:**

**Summary:** The authors propose a method for augmenting AV datasets with potentially dangerous maneuvers. The dangerous scenes are generated based on hand-designed primitives using image synthesis techniques and produce novel scenes. The authors evaluate the generated scenes on corner case score evaluation and a human study to argue that scenes are realistic and depict dangerous behaviors.

**Somewhat limited originality:** The key contribution of the paper is the generation of anomalous/dangerous scenarios based on hand-designed primitives. The authors leverage existing tools for a majority of the scene generation synthesis and analysis, but do contribute a set of motion primitives which can be used to augment existing data. The primitives are based on dangerous driving behavior one may observe from a human and well motivated. The idea is certainly promising but could be improved with a larger or automatically synthesized library of primitives.

**Difficult to asses significance:** The results are somewhat incomplete and have a handful of issues.
- The method for evaluating corner case score by determining a measure of image non-predictability seems like a strong method for measuring atypicality/how anomalous a scene is. However, an anomalous scene/corner case is not necessarily dangerous. It’s possible that the scenarios are difficult to for the prediction module to accurately predict but not necessarily more dangerous/hazardous.
- In addition, I would argue that that the corner case score is not a good proxy for danger because of the results in the slalom lane change column of Table 1. The increase in corner score due to the slalom primitive is minimal (and potentially within the variance — which the authors do not report). The minimal increase is possibly because the vehicle oscillates about a centered-in-lane prediction. However, I would consider this very risk behavior. I see a similar issue with the exit parking column in Table 1.
- In the user study, the authors do not present realistic/anomaly/novelty/risky scores for standard scenarios which have not been augmented with DANGER. This seems like a major weakness, and further because the scores in Fig 3 have high variance. If the users rated standard driving scenarios as 10 on the realistic scale, then it would incidate that the augmentation is not very realistic, in contrast to the authors’ claim on L305-L306.

**Quality / Clarity:** The authors are clear about the techniques they use and develop. The primitives are well motivated and explained and the appendix adds a nice amount of detail about the dangerous behaviors the authors are considering in the primitives. The legend and axes in Figure 2 are very difficult to read and the axes in Figure 3 are also difficult to read.


**Issues:**
- The authors don’t demonstrate that the augmentations improve the robustness of an autonomous vehicle dataset, which they suggest as key use for DANGER (L45-L46, L314-L315). The authors would need to show that a method trained on, for example, DANGER-vKITTI versus vKITTI results in some improvement in safety or even just detection of unsafe scenarios.
- The authors do not include the instructions or questions given to users in the user study. This seems necessary as questions can bias the users towards a specific result. This is doubly important because there is no control (i.e. a non-augmented scenario)  in the study. I cannot accept the authors’ interpretation of the user study without at least a control for comparison and ideally seeing the questions asked.
- I do not think that the corner case score is a reliable proxy for danger because of the results in the slalom lane change in Table 1, see comments on the significance above.
- L309-L310: Without evidence, it seems suspect to argue that subjects didn’t view the braking scenario as risky because they didn’t watch the scenario for long enough. Perhaps the users didn’t believe the scenarios were as risky because heavy braking is a common behavior, or it is difficult to tell that the car is breaking heavily. To me, it not even clear which vehicle is supposed to be dangerous in Figure 17.

---

### Official Review · Reviewer_cuT6 · 2022-10-16
**Synthetic Dangerous Scenes for Testing Autonomous Driving**

**Rating:** 5
**Confidence:** 4

**Review:**

In this work, the authors propose a framework to generate dangerous scenes for testing autonomous driving systems with predefined primitives. The authors demonstrated the generated scenarios/images to be realistic. Finally, the authors conduct a human study to evaluate the generated scenarios/images. While this work is quite complete, there are several key limitations.

First, the motivation is less clear. Given a synthetic dataset and an existing renderer, the motivation for modifying the images from the dataset instead of simulating new scenarios is unclear to the authors. Modifying images from the real-world dataset would make more sense to the review.

Second, the proposed method is with limited novelties. The key ingredients here are the renderer and the scenarios primitives. However, the authors directly used an existing renderer and the primitives are manually constructed.

Also, the submitted manuscript is over the limitation of 9 pages.

---

### Official Review · Reviewer_nJbm · 2022-10-17
**Scene editing framework to generate dangerous driving scenarios for stress testing AV stacks**

**Rating:** 8
**Confidence:** 4

**Review:**

This paper presents an approach for generating dangerous driving scenarios by editing vanilla driving scenarios in AV datasets. The efficacy of the approach is validated by a user study with 100 participants.

The lack of safety-critical scenarios in AV datasets is a major impediment to building safe and trustworthy AVs. The framework presented in this paper, DANGER, is an important step towards addressing this important issue. Overall, I consider this to be a worthy contribution to the workshop. I only have a couple of minor comments:
1. Do the authors have more details on the realism of the scene editing?
2. To my understanding, DANGER is a platform for scene editing, but how to edit the scene is still not an automated process. I would appreciate any insights on the challenges in automated scene editing to generate realistic dangerous scenarios.

---

### Decision · Program_Chairs · 2022-10-21

**Decision:**

Accept

**Comment:**

The reviewers have mixed attitudes toward this paper. The paper focuses on an important question that is closely related to the subject of the workshop. While the paper is quite complete, the reviewers pointed out several issues of this paper. The main ones are the lack of novelty and clarify. We eventually decided to accept this paper as it studied a very important problem for safe and trustworthy AVs. It will be of interest to the target audience of the workshop such as the first reviewer who voted strong accept. However, please try to address the reviewers' concerns in the final version as much as possible to improve the quality of the paper.